# ALADA Dose Optimization in the Computed Tomography of the Temporal Bone: The Diagnostic Potential of Different Low-Dose CT Protocols

**DOI:** 10.3390/diagnostics11101894

**Published:** 2021-10-15

**Authors:** Barbara Kofler, Laura Jenetten, Annette Runge, Gerald Degenhart, Natalie Fischer, Romed Hörmann, Michael Steurer, Gerlig Widmann

**Affiliations:** 1Department of Otorhinolaryngology, Medical University of Innsbruck, Anichstrasse 35, 6020 Innsbruck, Austria; ba.kofler@tirol-kliniken.at (B.K.); natalie.fischer@tirol-kliniken.at (N.F.); 2Department of Radiology, Medical University of Innsbruck, Anichstrasse 35, 6020 Innsbruck, Austria; laura.jenetten@tirol-kliniken.at (L.J.); gerald.degenhart@tirol-kliniken.at (G.D.); m.steurer@tirol-kliniken.at (M.S.); 3Division of Clinical and Functional Anatomy, Medical University of Innsbruck, Müllerstrasse 59, 6020 Innsbruck, Austria; romed.hoermann@i-med.ac.at

**Keywords:** X-ray computed tomography, spiral cone-beam computed tomography, radiation exposure, temporal bone, middle ear, inner ear

## Abstract

Objective: Repeated computed tomography (CT) is essential for diagnosis, surgical planning and follow-up in patients with middle and inner ear pathology. Dose reduction to “as low as diagnostically acceptable” (ALADA) is preferable but challenging. We aimed to compare the diagnostic quality of images of subtle temporal bone structures produced with low doses (LD) and reference protocols (RP). Methods: Two formalin-fixed human cadaver heads were scanned using a 64-slice CT scanner and cone-beam CT (CBCT). The protocols were: RP (120 kV, 250 mA, CTDIvol 83.72 mGy), LD1 (100 kV, 80 mA, CTDIvol 26.79 mGy), LD2 (100 kV, 35 mA, CTDIvol 7.66 mGy), LD3 (80 kV, 40 mA, CTDIvol 4.82 mGy), and CBCT standard protocol. Temporal bone structures were assessed using a 5-point scale. Results: A median score of ≥2 was achieved with protocols such as the tendons of m. tensor tympani (RP/LD1/LD2/CBCT) and m. stapedius (CBCT), the incudostapedial joint (RP/LD1/CBCT), the incudomalleolar joint (RP/LD1/LD2/CBCT), the stapes feet (RP/LD1/CBCT), the stapes head (RP/LD1/LD2/CBCT), the tympanic membrane (RP/LD1/LD2/CBCT), the lamina spiralis ossea (none), the chorda tympani (RP/LD1/CBCT), and the modiolus (RP/LD1/LD2/CBCT). Adaptive statistical iterative reconstructions did not show advantages over the filtered back projection. Conclusions: LD protocols using a CTDIvol of 7.66 mGy may be sufficient for the identification of temporal bone structures.

## 1. Introduction

Multi-slice computed tomography (CT) imaging is an essential part of the diagnostic workup for temporal bone disease. Pathologies of the middle and inner ear, such as malformations, inflammation, trauma and especially cancer can be assessed precisely for preoperative planning [1]. High positive and negative predictive values of up to 91% for intraoperative findings involving middle ear structures—i.e., ossicles or the round window—as well as the extent of cholesteatoma and neoplasms have been described in earlier studies on high-resolution CTs of the temporal bone [2,3,4,5]. Otologic surgeons, therefore, prefer high-resolution CTs to avoid image noise and the blurred visualization of subtle temporal bone structures.

Despite these beneficial aspects, high-resolution CT scans harbor a risk for radiation-induced cancer and the development of cataracts. Radiation exposure during childhood [6,7,8] and in the reproductive age [9] can cause serious genetic damage. The radiation doses patients are exposed to in modern helical scanners are defined by the CT dose index volume (CTDI_vol_). The CTDI_vol_ during a high-resolution CT of the temporal bone can be as high as 84 mGy, which is 7 times more than a thoracoabdominal CT scan. In addition to the scanned body part, the absorbed radiation dose depends on patient factors such as age and body mass. The effective radiation dose of a temporal bone CT scan is almost four times higher in a one-year-old child than in an adult [10]. According to Australian Medicare records, the incidence of cancer was 24% higher in children who had received a CT scan than in those who had not been exposed to CT radiation. Furthermore the correlation between radiation doses and the incidence of cancer, the “incidence rate ratio”, increased with each additional CT scan, especially at a younger age [11]. In 2776 patients undergoing CT for the staging of head and neck neoplasms, repeated radiation exposure was also associated with a higher risk of cataracts [12].

Fortunately, modern CT technology allows for low-dose (LD) exposures and short scanning times. As a consequence, the reduction of radiation exposure to the level of “as low as *reasonably achievable*” (ALARA) has become a major task in radiology [13]. Briefly, the ALARA principle comprises measures to reduce exposure (i.e., scanning) time, enhance the distance to the radioactive source and to shield with appropriate materials. Appropriate standard reference values have since been implemented by national medical and legislative authorities [14,15,16]. As a “refinement,” the principle of “as low as *diagnostically acceptable*” (ALADA) was initially introduced for cone-beam CTs (CBCT) of the maxillary and dental regions. ALADA combines the ALARA principle with the appropriate settings of tube current time product (mAs), kilovolt peak (kVp), and high-definition and -resolution parameters, depending on the indication for imaging [17]. The resulting images of dental and bony structures have been reported to be diagnostically acceptable and interpretable even at dose reductions down to one-eighth of the manufacturer’s recommended standard value [18].

At the Department of Radiology at the Medical University of Innsbruck, diagnostic reference levels were implemented for CT as recommended by European societies [19]. However, for the sole purpose of identifying minuscule temporal bone structures in the middle or inner ear, low-dose protocols may suffice when iterative reconstructive techniques (IRT) are applied to compensate for increases in image noise. In light of scarce data on the application of the ALADA principle in temporal bone imaging, we aimed for a translational approach. The purpose of this study was to evaluate the diagnostic image quality of subtle temporal bone structures with an aim to identify ALADA protocols for a 64-slice CT scanner.

## 2. Materials and Methods

In this study, two cadaver heads prepared by the Department of Clinical and Functional Anatomy were used. The anonymous donors had given their informed consent for use for scientific purposes prior to death [20,21]. The cadaver heads had been preserved by arterial injections of a formaldehyde-phenol solution and an alcohol-glycerin solution, as well as immersion in phenolic acid in water for 1–3 months [22].

### 2.1. CT Scanning

The cadaver heads were scanned using a 64-slice CT scanner (Discovery CT750 HD, GE Healthcare, Vienna, Austria). The standard protocol for temporal bone CT imaging was used. In addition to the standard protocol, a subsequent series of LD protocols with reduced kV and mAs were executed (see Table 1). All images were reconstructed using filtered back projection (FBP) to minimize blurring. For noise reduction, the images were reconstructed using specific algorithms and the vendor-specific IRT, namely adaptive statistical iterative reconstruction 50 (ASIR 50) and ASIR 100 (see Table 1).

### 2.2. CBCT Scanning

Both cadaver heads were also scanned using a CBCT scanner (KaVo 3D eXam, KaVo Dental GmbH, Biberach, Riß, Germany) with the following protocol: 120 kV, 5 mAs, and 0.2 mm slice thickness.

### 2.3. Dose Estimation

CTDI_vol_ and dose-length product (DLP) were recorded from the (DICOM) tags (see Table 1). The doses of the CT protocols are given in Table 2. The DLP was calculated for a scan length of 10 cm. The effective dose was calculated based on the conversion factors from DLP to the effective dose as a function of voltage, region, and age for ICRP Publication 103, according to Deak el al. [10].

### 2.4. Analysis of Diagnostic Image Quality

Each image series was transferred to the DICOM format using the IMPAX EE picture archiving and communication system (PACS; Agfa HealthCare, Bonn, Germany). The images were assessed by three examiners, including one head and neck radiologist at the senior consultant level and two otolaryngologists who were board-examined specialists. The images were blinded for imaging mode, protocol and possible preexisting ear pathologies. Images were analyzed using IMPAX EE application software (Agfa HealthCare, Bonn, Germany) and high-resolution diagnostic color LCD monitors (Totoku CCL254i, Totoku Europe GmbH, Rein Medical GmbH, Willich, Anrath, Germany). The extended MPR plugin was applied for image orientation. Axial images were oriented parallel to the infraorbitomeatal line, coronal images were oriented perpendicularly to this plane, and stapes-MPRs were oriented parallel to the long axis of the stapes.

The following ten anatomical structures were assessed using a 5-point scale similar to one applied by Pein and coworkers [23]: 1—not visible, 2—faintly visible, 3—visible, 4—well visible, and 5—very well visible. The visibility of the following structures was rated: the tendon of the m. tensor tympani, the tendon of the m. stapedius, the incudostapedial joint, the incudomalleolar joint, the stapes feet, the stapes head, the tympanic membrane, the lamina spiralis ossea, the chorda tympani, and the modiolus. In addition, we described whether the selected structures were shaded and, thus, difficult to discern or not visible at all. Shadowing was caused by fluid deposits in the pneumatized spaces of the preserved cadaver heads at the time of image data acquisition. The cadaver heads were not moved between the different CT protocols, and all CT scans were performed during one session. Thus, shading patterns were identical independent of the applied CT protocol. However, the shading patterns of CT and CBCT scans varied because the cadaver heads had to be moved to a different device (Table 3).

### 2.5. Statistical analyses

PASW statistics (version 15.0, SPSS) was used for data analyses and descriptive statistics. Continuous data was shown as mean ± standard deviation (SD). The Wilcoxon signed-rank test was used to evaluate significant differences in visibility scores depending on imaging modes (CT vs. CBCT), dose protocols (RP vs. LD1-LD3) and the effect of IRT. A *p*-value of ≤0.05 was considered statistically significant. The null hypothesis proposed that dose reductions and IRT do not change diagnostic image quality compared to the reference dose protocol (RP).

## 3. Results

Different LD protocols (LD1, LD2, and LD3) of CT and CBCT images of the temporal bone were assessed using the 5-point scale and compared with the standard protocol. The visibility of anatomical structures was rated >2 (visible, well visible, and very well visible) on the visual grading scale when the following protocols were applied (protocols/imaging device in brackets): the tendon of the m. tensor tympani (RP/LD1/LD2/CBCT), the tendon of the m. stapedius (CBCT), the incudostapedial joint (RP/LD1/LD2/CBCT), the incudomalleolar joint (RP/LD1/LD2/LD3/CBCT), the stapes feet (RP/LD1/CBCT), the stapes head (RP/LD1/LD2/CBCT), the tympanic membrane (RP/LD1/LD2//CBCT), the lamina spiralis ossea (none), the chorda tympani (RP/LD1/CBCT), and the modiolus (RP/LD1/LD2/CBCT). The application of both ASIR 50 and ASIR 100 did not yield higher visibility scores.

### 3.1. CT

There were no significant differences regarding the visualization of the majority of anatomical structures after the application of the RP and the LD1 protocols, except for the stapes head and the incudomalleolar joint (see Figure 1 and Figure 2, and Table 4). There were no significant differences in the visibility of select anatomical structures after scanning according to the RP and LD2 protocols (the tendon of the m. stapedius, the stapes feet, and the lamina spiralis ossea, see Table 4). Visibility decreased significantly with LD3 protocol scans. In addition, the increased noise and faint visualization of the structures in the scans were generated by the application of LD3 protocols. ASIR 50 and 100 did not procure any changes in visibility scores at significant levels in the images of LD2 and LD3 protocols.

### 3.2. CBCT

There was no significant difference in the diagnostic visibility of most anatomical structures in CBCT in comparison with CT RP (Figure 3). The visibility scores for the m. stapedius tendon were significantly higher upon the examination of images generated with CBCT than with the CT RP (Table 4).

## 4. Discussion

This study was designed to evaluate and compare the visibility of subtle middle ear structures using different LD CT protocols, as well as CBCT. ALADA protocols in temporal bone imaging produce diagnostically applicable images at a minimal dose [17,24,25]. CT is widely available and more cost-effective than other modes of cross-sectional imaging. In addition, there have been many refinements of CT and CBCT technology in recent years. Not only children with pathologies of the inner and middle ear, but also patients of all ages with other conditions in need of repeated CT examinations, can benefit from a dose reduction [11].

According to our results, there were only minor differences in the visualization of minuscule temporal bone structures upon the examination of images generated with the LD1 protocol, CBCT, and the CT RP. All but one structure (the lamina spiralis ossea) was visible with all three protocols. As comparable diagnostic value can be provided by the LD1 protocol and CBCT, both imaging modes may be regarded as equal alternatives to the RP.

Interestingly, the radiation dose in the LD1 protocol comprises only one-third of the dose applied in the CT RP. The results of several study groups reporting sufficient image quality at low radiation doses confirm our data. According to Tada and coworkers, the visualization of the middle and inner ear structures of pediatric patients was sufficient in LD 320-row CTs compared to standard protocol 320-row CT scans [26]. In addition to the adjustment of kVp or mAs, the radiation dose of a CT examination can be reduced by increasing the number of detection rows. Bauknecht and coworkers observed a comparable image quality in 320-row CT and 16-row CT scans of the temporal bone. However, radiation exposure was only decreased by one-sixth, whereas changes in mAs and kVp can result in dose reductions of up to 50% in temporal bone CTs [27,28]. However, all measures of dose reduction lead to increased image noise. In our study, the LD2 and LD3 protocols produced increased noise and artifacts. The reduction of image noise through IRT did not improve the visualization of anatomical structures. The strongest positive effect of IRT was seen in the RP. In the LD2 and LD3 protocols, the IRT did not show any statistically significant advantage. The LD3 protocol may not be sufficient for the diagnostic evaluation of subtle temporal bone structures. With regard to clinical application, optimized protocols in CTs with multiple rows can produce an optimal relation of dose reduction and visibility, such as an LD1 protocol in a 64-row scanner.

The visibility of subtle structures in CT and CBCT were also compared. The visibilities of the tensor tympani muscle, the incudostapedial joint, the stapes feet, the tympanic membrane, the chorda tympani, and the modiolus were equal in CBCT and CT. The visibility of the stapedius muscle tendon was rated even better in CBCT than in RP CT or any LD protocol. Pein and coworkers reported the visualization of middle ear structures in 38 patients to be slightly better in CBCT images than in an optimized standard protocol CT, especially with regard to the stapes structures (the stapes crura and the tendon of the stapedius muscle, *p* = 0.003 and *p* = 0.033, respectively). Inner ear structures such as the modiolus and the lamina spiralis ossea were more clearly defined in CT images with optimized protocols (*p* = 0.001). These authors concluded that optimized CT and CBCT protocols were a prerequisite for the equal visualization of temporal bone structures [23]. Interestingly, the lamina spiralis ossea was not visible in any imaging mode in this study. This structure is a thin, two-layered, bony, three-dimensional winding helix separating the scala tympani and vestibuli. The identification of the lamina spiralis ossea requires an experienced observer even in high-resolution CTs, which should be reserved for highly specific purposes.

CT and CBCT have been directly compared in many studies, resulting in contradictory results in terms of resolution and radiation dose [29,30,31,32,33]. Due to the inhomogeneous image quality of different scanners and increased susceptibility to motion artifacts, CBCT should not be generally seen as a replacement for the CT [23]. Radiation exposure is lower in CBCT in the high-contrast range and it was described to be superior to CT in terms of the resolution of high-contrast structures [34]. A supine position is not always necessary in CBCT exams, which is a practical aspect in the imaging of uncooperative patients. Other studies suggested CT to be the preferable imaging procedure. Consistently good image quality with fewer artifacts, higher soft tissue resolution, and short scan times were considered advantageous [32,35]. A CT scan can be carried out in the unconscious patient and emergency situations, i.e., a polytrauma. In CBCT, a small subunit of the body, such as a single temporal bone, can be visualized, whereas in CT the whole planar section has to be scanned. The selective examination of only one temporal bone reduces the required radiation dose to one-third of the dose a patient is exposed to in a CT study [34]. On the other hand, scanning both sides may be beneficial for discerning individual anatomical variations and pathologies by comparing symmetry. Furthermore, the eye lens is in the primary radiation field of CBCT. Due to gantry tilting, this exposure can be avoided in CT [23].

LD protocols may be an option in patients requiring repeated CT scans, as even changes in very subtle temporal bone structures are recognizable. Adequate diagnostic imaging for follow-up during intracranial procedures or the planning of surgery in children with craniofacial anomalies with LD CTs protocols was feasible in other studies [36,37]. Craniofacial anomalies often go along with middle ear deformities or chronic otitis media. Preoperative planning and postoperative follow-up can be simplified by LD CT protocols with an adequate visualization of middle ear structures and may, in some cases, render additional surgeries unnecessary. Cumulative doses in children with spinal or cardiac anomalies were reported to be as high as 23 mSv on average, with an associated lifetime attributable risk of cancer up to 6.5% [38,39]. In light of this, the possibility to reduce the effective dose to roughly a fourth (LD1) or a tenth (LD2) of the RP dose cannot be pointed out often enough. Whenever applicable, LD protocols should be considered when repeated CT scans are unavoidable in the pediatric population. In addition, sedation can be avoided when CTs are applied in young children instead of MRI. In clinical practice, repeated scanning is mostly applied for the monitoring of disease courses in larger structures (i.e., the lung). Infiltrates, pattern changes, and the growth of masses are evaluated but not every tiny aspect of the organ is relevant. As an example, LD protocols have been established as a screening method for early-stage lung cancer with positive effects seen on mortality rates. However, there is a limit to the reduction of radiation dose in temporal bone CT. Only the comparatively large incudomalleolar joint was visible in LD3 protocol scans of this study, whereas all other structures were heavily shaded. In the confined space of the middle ear with submillimetric structures, a higher resolution is required than in visceral organs. IRT cannot reduce image noise sufficiently in ultra-low-dose CT images of the temporal bone.

Studies explicitly focusing on the realization of the ALADA principle are scarce. However, most authors seem to agree on the fact that IRT was essential to guarantee diagnostic value. In general, diagnostically acceptable LD CT scans of large structures such as the pelvis, spine and lung were produced with a CTDI_vol_ as low as 0.9 mGy. As the effective dose could be reduced to 6% of the routine dose, ALADA protocols were even deemed acceptable as a diagnostic tool in pediatric lung disorders [40,41,42]. With the application of iterative reconstructive techniques, LD dynamic myocardial CT perfusion was feasible without an additional reduction of the myocardial blood flow [43].

This study has several limitations. First, all imaging studies were performed with only two cadavers. Furthermore, due to the method of preservation and the storage of the skulls, there were liquid deposits in the pneumatized spaces and partly in the middle ear. Some structures became opacified and unrecognizable. Thus, the visibility may have been rated worse than it would have been in a fresh cadaver or living subject without opacification. On the other hand, liquid deposits and erosions in the middle ear simulate the opacification seen in otitis media, which may increase the clinical value of our study.

Future work should define phantom-based qualitative reference quality parameters such as spatial resolution, or contrast-to-noise ratios of clinically proven ALADA protocols, which may be used to define ALADA doses for different manufacturers or scanner models.

## 5. Conclusions

For CT of the temporal bone, only minor differences in the diagnostic visualization of anatomical structures were found between the RP (CTDI_vol_ 83.72 mGy), the LD1 protocol (CTDI_vol_ of 26.79 mGy), and CBCT. The appropriate imaging mode should be selected according to the diagnostic problem at hand with regard to the cooperation and age of the patient. If temporal bone CTs in children are necessary, the LD1 protocol may provide a sensible option due to the significant reduction of the effective dose. Further dose reductions cannot be recommended according to our data because of significant decreases in visibility, which cannot be improved by IRT. In general, ALADA protocols should be selected for temporal bone imaging whenever clinically acceptable.

## Figures and Tables

**Figure 1 diagnostics-11-01894-f001:**
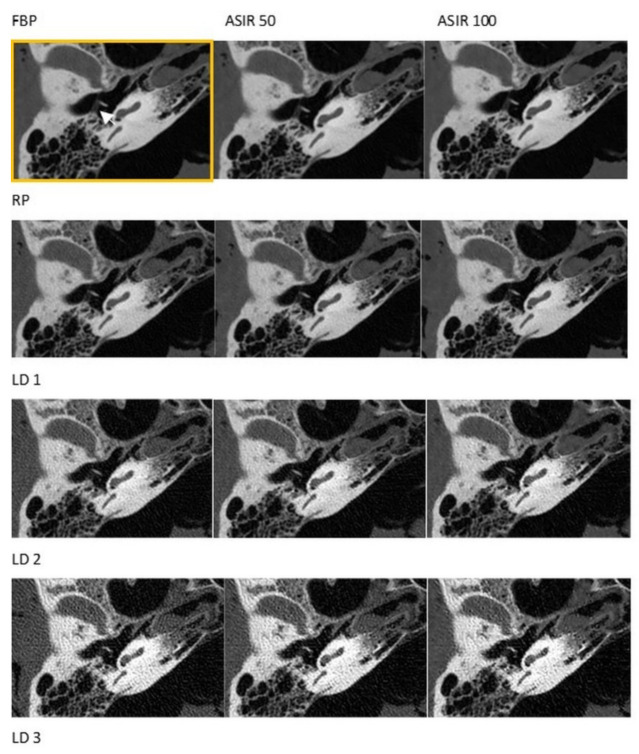
Visualization of the chorda tympani (white arrow). RP = reference protocol (83.72 mGy), LD1 = low dose 1 protocol (26.79 mGy), LD2 = low dose 2 protocol (7.66 mGy), LD3 = low dose 3 protocol (4.82 mGy). Yellow box = reference image using reference protocol and filtered back projection.

**Figure 2 diagnostics-11-01894-f002:**
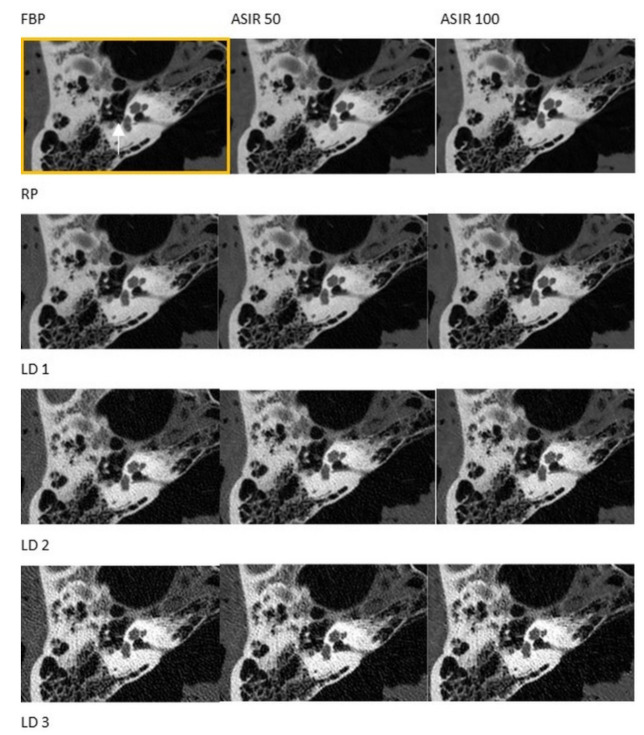
Visualization of the stapes (white arrow). RP = reference protocol (83.72 mGy), LD1 = low dose 1 protocol (26.79 mGy), LD2 = low dose 2 protocol (7.66 mGy), LD3 = low dose 3 protocol (4.82 mGy). Yellow box = Reference image using reference protocol and filtered back projection.

**Figure 3 diagnostics-11-01894-f003:**
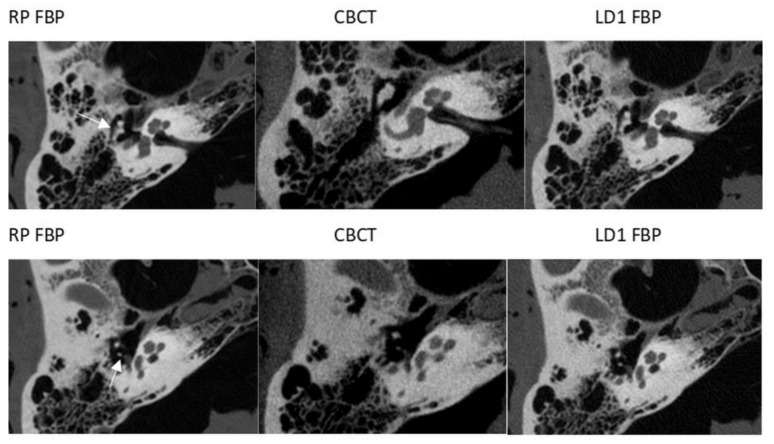
Visualization of the incudomalleolar joint (upper row) and incudostapedial joint (lower row) (white arrows). RP = reference protocol (83.72 mGy), FBP = filtered back projection, CBCT = cone-beam CT, LD1 = low dose 1 protocol (26.79 mGy).

**Table 1 diagnostics-11-01894-t001:** Technical data of the multi-slice CT protocols.

Unit	Measurements
kV	120
mAs	250
Table speed/Table feed per rotation	10,625/10,625
Pitch (mm)	0.531
Collimation	20 × 10,625
Slice thickness (mm)	0.625
Recon increment	0.625
FOV, Scan length	individual (as small as possible)
Kernel	BONE2
CTDIvol (reference) (mGy)	83.72
CTDIvol (120 kV/80 mAs) = LD1 (mGy)	26.79
CTDIvol (100 kV/35 mAs) = LD2 (mGy)	7.66
CTDIvol (80 kV/40 mAs) = LD3 (mGy)	4.82
FBP/Kernel	FBP/BONE2
IRT/Kernel	ASIR 50/BONE2
IRT/Kernel	ASIR 100/BONE2

kV = kilovolt, mAs = milliampere second, FOV = field of view, CTDI = computed tomography dose index, FBP = filtered back projection, IRT = iterative reconstruction technology.

**Table 2 diagnostics-11-01894-t002:** Conversion factors from DLP to effective dose as a function of voltage, region, and age as per ICRP Publication 103. 2007 Recommendations of the International Commission on Radiological Protection. DLP = dose length product.

Protocol	RP	LD1	LD2	LD3
kV/mA	120/250	120/80	100/35	80/40
Scanlength (cm)	10	10	10	10
DLP (mGy × cm)	837.2	267.9	76.6	48.2
Conversion factor adult	0.0019	0.0019	0.0019	0.0018
Effective dose adult (mSv)	1.59068	0.50901	0.14554	0.08676
Conversion factor 10-year-old (mSv)	0.0027	0.0027	0.0027	0.0026
Effective dose 10-year-old (mSv)	2.26044	0.72333	0.20682	0.12532
Conversion factor 5-year-old	0.0035	0.0035	0.0035	0.0035
Effective dose 5-year-old (mSv)	2.9302	0.93765	0.2681	0.1687
Conversion factor 1-year-old	0.0053	0.0053	0.0054	0.0056
Effective dose 1-year-old (mSv)	4.43716	1.41987	0.41364	0.26992
Conversion factor newborn	0.0085	0.0085	0.0088	0.0094
Effective dose newborn (mSv)	7.1162	2.27715	0.67408	0.45308

**Table 3 diagnostics-11-01894-t003:** Shading patterns depending on imaging device.

Imaging Device	Cadaver Head	Side	Shaded Structure
CT	Number 1	Right	None
		Left	None
	Number 2	Right	Tendon of the m. tensor tympani Tendon of the m. stapedius Tympanic membrane Chorda tympani Stapes
		Left	Stapes
CBCT	Number 1	Right	None
		Left	None
	Number 2	Right	None
		Left	Tympanic membrane Chorda tympani Modiolus Stapes

CT = computed tomography, CBCT = cone-beam CT.

**Table 4 diagnostics-11-01894-t004:** Median score and range (min.–max.).

	Protocol	Score FBP	*p*-Value	Score ASIR 50	*p*-Value	Score ASIR 100	*p*-Value
Tendon of m. tensor tympani	RP LD1 LD2 LD3 CBCT	4.0 (1–5) 3.0 (1–4) 2.5 (1–3) 2.0 (1–3) 3.0 (1–4)	n/a 0.083 0.023 * 0.023 * 0.336	4.0 (1–5) 3.0 (1–4) 2.5 (1–3) 2.0 (1–3) 3.0 (1–4)	n/a 0.083 0.023 * 0.023 * 0.336	4.0 (1–5) 3.0 (1–4) 3.0 (1–3) 2.0 (1–3) 3.0 (1–4)	n/a 0.046 * 0.023 * 0.026 * 0.236
Tendon of m. stapedius	RP LD1 LD2 LD3 CBCT	2.0 (1–3) 1.5 (1–3) 1.0 (1–2) 1.0 (1–1) 2.5 (1–4)	n/a 0.157 0.059 0.046 * 0.046 *	2.0 (1–3) 1.5 (1–3) 1.0 (1–2) 1.0 (1–1) 2.5 (1–4)	n/a 0.157 0.059 0.046 * 0.046 *	2.0 (1–3) 1.5 (1–3) 1.0 (1–2) 1.0 (1–1) 2.5 (1–4)	n/a 0.157 0.059 0.046 * 0.046 *
Incudostapedial joint	RP LD1 LD2 LD3 CBCT	4.0 (3–4) 3.5 (3–4) 3.0 (2–3) 2.0 (1–3) 3.5 (3–4)	n/a 0.083 0.008 * 0.011 * 0.083	4.0 (3–4) 3.5 (3–4) 3.0 (2–3) 2.0 (1–3) 3.5 (3–4)	n/a 0.083 0.008 * 0.011 * 0.083	4.0 (3–5) 3.5 (3–4) 3.0 (2–3) 2.0 (1–3) 3.5 (3–4)	n/a 0.046 * 0.009 * 0.011 * 0.046 *
Incudomalleolar joint	RP LD1 LD2 LD3 CBCT	5.0 (4–5) 4.0 (4–4) 3.5 (3–4) 3.0 (1–4) 4.0 (3–4)	n/a 0.014 * 0.015 * 0.017 * 0.023 *	5.0 (4–5) 4.0 (4–5) 3.5 (3–4) 3.0 (1–4) 4.0 (3–4)	n/a 0.025 * 0.015 * 0.017 * 0.023 *	5.0 (4–5) 4.0 (4–5) 3.5 (3–4) 3.0 (1–4) 4.0 (3–4)	n/a 0.046 * 0.015 * 0.017 * 0.023 *
Stapes feet	RP LD1 LD2 LD3 CBCT	3.0 (1–4) 2.5 (1–4) 1.5 (1–3) 1.5 (1–2) 3.0 (2–3)	n/a 0.083 0.059 0.024 * 0.750	3.0 (1–4) 2.5 (1–4) 1.5 (1–3) 1.5 (1–2) 3.0 (2–3)	n/a 0.083 0.059 0.024 * 0.750	3.0 (1–5) 2.5 (1–4) 1.5 (1–3) 1.5 (1–2) 3.0 (2–3)	n/a 0.046 * 0.066 0.026 * 1.000
Stapes head	RP LD1 LD2 LD3 CBCT	4.0 (3–4) 3.0 (2–4) 2.5 (1–3) 2.0 (1–2) 3.0 (3–4)	n/a 0.014 * 0.015 * 0.008 * 0.046 *	4.0 (3–4) 3.0 (2–4) 2.5 (1–3) 2.0 (1–2) 3.0 (3–4)	n/a 0.014 * 0.015 * 0.008 * 0.046 *	4.0 (3–4) 3.0 (2–4) 2.5 (1–3) 2.0 (1–2) 3.0 (3–4)	n/a 0.014 * 0.015 * 0.008 * 0.046 *
Tympanic membrane	RP LD1 LD2 LD3 CBCT	4.0 (1–5) 3.0 (1–5) 2.5 (1–4) 2.0 (1–3) 3.0 (1–4)	n/a 0.157 0.020 * 0.014 * 0.206	4.0 (1–5) 3.0 (1–5) 2.5 (1–4) 2.0 (1–3) 3.0 (1–4)	n/a 0.157 0.020 * 0.014 * 0.206	4.0 (1–5) 3.0 (1–5) 2.5 (1–4) 2.0 (1–3) 3.0 (1–4)	n/a 0.157 0.020 * 0.014 * 0.206
Lamina spiralis ossea	RP LD1 LD2 LD3 CBCT	2.0 (2–2) 2.0 (2–2) 2.0 (1–2) 1.5 (1–2) 2.0 (2–2)	n/a 1.000 0.157 0.046 * 1.000	2.0 (2–2) 2.0 (2–2) 2.0 (1–2) 1.5 (1–2) 2.0 (2–2)	n/a 1.000 0.157 0.046 * 1.000	2.0 (2–2) 2.0 (2–2) 2.0 (1–2) 1.5 (1–2) 2.0 (2–2)	n/a 1.000 0.157 0.046 * 1.000
Chorda tympani	RP LD1 LD2 LD3 CBCT	3.0 (1–4) 3.0 (1–3) 2.0 (1–2) 1.0 (1–2) 3.0 (1–3)	n/a 0.083 0.024 * 0.026 * 1.000	3.0 (1–4) 3.0 (1–3) 2.0 (1–2) 1.0 (1–2) 3.0 (1–3)	n/a 0.083 0.024 * 0.026 * 1.000	3.0 (1–4) 3.0 (1–3) 2.0 (1–2) 1.0 (1–2) 3.0 (1–3)	n/a 0.083 0.024 * 0.026 * 1.000
Modiolus	RP LD1 LD2 LD3 CBCT	4.0 (3–5) 3.5 (3–5) 3.0 (2–4) 3.0 (2–3) 3.0 (3–4)	n/a 0.317 0.025 * 0.014 * 0.083	4.0 (3–5) 3.5 (3–5) 3.0 (2–4) 3.0 (2–3) 3.0 (3–4)	n/a 0.317 0.025 * 0.014 * 0.046 *	4.0 (3–5) 3.5 (3–5) 3.0 (2–4) 3.0 (2–3) 3.0 (3–4)	n/a 0.317 0.025 * 0.014 * 0.046 *

Scoring and statistical comparison with RP. RP = reference protocol (83.72 mGy), LD1 = low dose 1 protocol (26.79 mGy), LD2 = low dose 2 protocol (7.66 mGy), LD3 = low dose 3 protocol (4.82 mGy). FBP = filtered back projection, * = statistically significant results (*p* ≤ 0.05). The columns labeled “score” refer to the applied 5-point scale used for grading visibility. n/a = *p*-value not available.

## Data Availability

Data supporting the reported results can be requested by emailing the corresponding author.

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
