# Peer review of "ALADA Dose Optimization in the Computed Tomography of the Temporal Bone: The Diagnostic Potential of Different Low-Dose CT Protocols"

_diagnostics, 2021, doi:10.3390/diagnostics11101894_

Round 1
Reviewer 1 Report
We can’t consider this article as an original article on diagnostic accuracy because the authors enrolled only two patients.
Lines 31 to 33
Multi-slice computed tomography (CT) imaging is an essential part of the diagnostic 31
workup for temporal bone disease . Pathologies of the middle and inner ear, such as 32
malformations, inflammatory processes, and trauma, can be assessed precisely for 33
preoperative planning and surgical outcome, expecially in cancer
The authors should cite
Cristalli G, Manciocco V, Pichi B, Marucci L, Arcangeli G, Telera S, Spriano G. Treatment and outcome of advanced external auditory canal and middle ear squamous cell carcinoma. J Craniofac Surg. 2009 May;20(3):816-21. doi: 10.1097/SCS.0b013e3181a14b99. PMID
Line 45-46
Despite the beneficial aspects, CT scans also harbor a risk for radiation-induced 45
cancer and the development of cataracts, especially after childhood exposure [4-6] and in the reproductive age
The authors could cite
Giannitto C, Campoleoni M, Maccagnoni S, Angileri AS, Grimaldi MC, Giannitto N, De Piano F, Ancona E, Biondetti PR, Esposito AA. Unindicated multiphase CT scans in non-traumatic abdominal emergencies for women of reproductive age: a significant source of unnecessary exposure. Radiol Med. 2018
Lines from 92 to 94
The image series were transferred to DICOM format using the IMPAX EE Picture 92
Archiving and Communication System (PACS; Agfa HealthCare, Bonn, Germany). The 93
images were assessed by three examiners, including one head and neck radiologist and 94
two otolaryngologists.
The authors should report the expertise of radiologists and otolaryngologists.
Are They blinded to histology results?
Lines from 116 to 117
They should introduce the differences of CT dose in patients.
Author Response
Dear Reviewer 1,
thank you very much for your valuable and highly appreciated comments on our work. I marked all your points in the comment section and am very happy to elaborate on any further questions.
with kind regards!
Reviewer 2 Report
This paper study the ALADA Dose Optimization in Computed Tomography of the Temporal Bone. There are a few weaknesses that should be addressed in this paper. Therefore, I suggest the authors resubmit it after a major revision. My suggestions are as follows: As the first step, I strongly suggest that the paper be proofread and reread meticulously again, particularly in regard to the spelling and grammatical mistakes. 1) Authors should enrich the literature review by addressing more relevant papers and recent papers. 2) Please divide the introduction into two parts literature review and introduction. 3) The limits of the results obtained in this paper are not mentioned. This point should be explained. 4) Comparisons with existing approaches are missing. 5) The conclusion is too short. 6) What is the purpose of the statistical analysis?Author Response
Dear Reviewer,
First of all thank you so much for your valuable contribution and comments on our work.
I have tried to comply with all your comments. There is one issue I'd like to elaborate on. I had had this manuscript proof read by a professional company. As an attachment you can find the certificate. Therefore I have not asked for proof reading service again this time.
All other sections are marked as comments. I'm eagerly awaiting further advice.
with kindest regards!

Round 2
Reviewer 1 Report
I think that this paper should be accepted as "Protocol" article type which presents a detailed introduction for proposed or ongoing clinical research, outlining the hyphothesis, rationale and methodology showing these preliminary results.If converted in " Protocol", this article could be accepted.
In my previous revision, I asked for CT doses of the two CT protocols that authors presented in Table two.
Author Response
Dear Reviewer 1,
thank you again so much for your highly valuable advice on our manuscript.
1. We changed the title according to your suggestion to:
ALADA dose optimization in computed tomography of the temporal bone- diagnostic potential of different low dose protocols
2. shading was rated during one CT session, as the cadaver head were not moved in between the different protocols. The shading patterns were thus recognized with the RP in both cadaver heads. CBCT was a different session, skulls had to be moved for that. I marked the according passages in the text.
We also added another table to present the effective radiation dose according to age, to underline the clinical value of the proposed LD protocols.
Thank you very mich for your help, with kind regards,
Annette Runge
Reviewer 2 Report
This version is available for publication.
Author Response
Dear Reviewer 2,
Thank you very much for your hihgly appreciated contribution. I did another spell check. I correspondence with my last and co corresponding author we alos added another table to underline the clinical value of LD protocols by pointing out the effective radiation dose depending on age.
with kind regards, Annette Runge